# Specificity of *Escherichia coli* Heat-Labile Enterotoxin Investigated by Single-Site Mutagenesis and Crystallography

**DOI:** 10.3390/ijms20030703

**Published:** 2019-02-06

**Authors:** Julie Elisabeth Heggelund, Joel Benjamin Heim, Gregor Bajc, Vesna Hodnik, Gregor Anderluh, Ute Krengel

**Affiliations:** 1Department of Chemistry, University of Oslo, Postbox 1033 Blindern, 0315 Oslo, Norway; j.e.heggelund@farmasi.uio.no (J.E.H.); j.b.heim@kjemi.uio.no (J.B.H.); 2Department of Biology, Biotechnical Faculty, University of Ljubljana, Jamnikarjeva 101, 1000 Ljubljana, Slovenia; gregor.bajc@bf.uni-lj.si (G.B.); vesna.hodnik@novartis.com (V.H.); 3Department of Molecular Biology and Nanobiotechnology, National Institute of Chemistry, Hajdrihova 19, 1000 Ljubljana; Slovenia; gregor.anderluh@ki.si

**Keywords:** bacterial toxin, cholera toxin, *Escherichia coli* heat-labile enterotoxin, lectin, *N*-acetyllactosamine binding, neutral glycosphingolipids, protein–carbohydrate interactions, surface plasmon resonance spectroscopy, X-ray crystal structure

## Abstract

Diarrhea caused by enterotoxigenic *Escherichia coli* (ETEC) is one of the leading causes of mortality in children under five years of age and is a great burden on developing countries. The major virulence factor of the bacterium is the heat-labile enterotoxin (LT), a close homologue of the cholera toxin. The toxins bind to carbohydrate receptors in the gastrointestinal tract, leading to toxin uptake and, ultimately, to severe diarrhea. Previously, LT from human- and porcine-infecting ETEC (hLT and pLT, respectively) were shown to have different carbohydrate-binding specificities, in particular with respect to *N*-acetyllactosamine-terminating glycosphingolipids. Here, we probed 11 single-residue variants of the heat-labile enterotoxin with surface plasmon resonance spectroscopy and compared the data to the parent toxins. In addition we present a 1.45 Å crystal structure of pLTB in complex with branched lacto-*N*-neohexaose (Galβ4GlcNAcβ6[Galβ4GlcNAcβ3]Galβ4Glc). The largest difference in binding specificity is caused by mutation of residue 94, which links the primary and secondary binding sites of the toxins. Residue 95 (and to a smaller extent also residues 7 and 18) also contribute, whereas residue 4 shows no effect on monovalent binding of the ligand and may rather be important for multivalent binding and avidity.

## 1. Introduction

The heat-labile enterotoxin (LT), a homologue of the cholera toxin (CT), is produced by enterotoxigenic *Escherichia coli* (ETEC). ETEC is responsible for millions of diarrheal cases and more than 50,000 deaths every year [1]. The mortality of the disease is declining, but the morbidity is not, despite improvements in sanitation facilities. ETEC infection in children often leads to long-term health problems like stunted growth and reduced cognitive abilities, triggering a vicious cycle of poverty [2]. The disease also affects travellers to endemic areas, including medical and military personnel, and has further been linked to chronic diseases like irritable bowel syndrome [3]. The infection spreads through the fecal–oral route, aggravated by the watery diarrhea caused by the enterotoxin.

LT and CT belong to the AB_5_ toxin family, consisting of one A-subunit and five B-subunits [4,5]. The A-subunit is catalytically active and is anchored to the center of the torus-shaped B-pentamer, which is responsible for the binding to epithelial cells. The B-pentamers of LT (LTB) and CT (CTB) share the same fold, and have approximately 80% sequence identity [5]. The binding to their main cellular receptor, the monosialoganglioside GM1, is one of the strongest carbohydrate–protein interactions known, with a binding constant of 43 nM [6,7]. The binding is enhanced by at least an order of magnitude when all five binding sites are occupied [8].

There are several crystal structures of CTB and LTB in complex with the GM1 pentasaccharide Galβ3GalNAcβ4[NeuAcα3]Galβ4Glc (Protein Data Bank (PDB) ID: 3CHB [9], 2XRQ [10], 1CT1 [11]). GM1 binds at the base of the B-pentamer, which faces the membrane, anchored by its two terminal residues: galactose and sialic acid (NeuAc). The galactose residue is buried in a deep cavity, whereas sialic acid binds mainly through water-mediated interactions at the protein surface.

Both LT and CT also bind other glycoconjugates at the cell surface, however, LT is generally more promiscuous and binds to a wider variety of glycosphingolipids than CT, including disialoganglioside GD2, asialo-GM1 and lacto-*N*-neotetraosylceramide (LNnT-Cer; paragloboside) [12,13] as well as intestinal polyglycosylceramides and sialic-acid containing glycosphingolipids [14]. Common for most of the additional purified ligands is that they contain a terminal galactose connected to either GalNAc or GlcNAc (Galβ3GalNAcβ4 or Galβ4GlcNAcβ3). There are two natural variants of LT, one isolated from human-infecting ETEC (hLT) and one from porcine-infecting ETEC (pLT). Although these two are very similar, they have different binding affinities. The B-pentamers hLTB and pLTB are identical except for four residues: 4, 13, 46, and 102, and of these, only residue 13 is situated at the primary binding site. Residue 13 is a histidine in most hLTB strains and an arginine in pLTB. A CT-like toxin from the bacterium *Citrobacter freundii* has also been shown to bind to LNnT-Cer, with an even higher affinity than pLTB [15]. This toxin has approximately 75% sequence identity to hLTB and CTB, and slightly lower sequence identity to pLTB.

In previous work, Teneberg and co-workers investigated the binding specificities of pLTB, hLTB, and CTB to two *N*-acetyllactosamine-terminated glycosphingolipids, branched lacto-*N*-neohexaosylceramide (LNnH-Cer, Galβ4GlcNAcβ6[Galβ4GlcNAcβ3]Galβ4GlcβCer) and linear lacto-*N*-neohexaosylceramide (Galβ4GlcNAcβ3Galβ4GlcNAcβ3Galβ4GlcβCer), immobilized to microtiter wells [16]. pLTB was shown to bind the strongest to both ligands, hLTB weaker, and CTB hardly at all. To test if the difference between hLTB and pLTB was due to residue 13, we subsequently generated the protein variant hLTB H13R, which showed enhanced binding to branched LNnH-Cer, strongly indicating that residue 13 is the key residue for this interaction [10]. However, the crystal structure of pLTB in complex with lacto-*N*-neotetraose (LNnT, Galβ4GlcNAcβ3Galβ4Glc; PDB ID: 2XRS [10]), shows no clear interaction between Arg13 and the ligand, suggesting that other residues are also important.

The differences between CTB and LTB were also investigated by cassette mutagenesis more than 20 years ago. Mutating residues 1–25 in concert with 94–95 in CTB gave rise to a toxin with binding specificities indistinguishable from hLTB, named LCTB*H* [14]. Changing residue Ser4 back to CTB-specific Asn4 generated toxin variant LCTB*K*, which showed strongly reduced binding to LNnT-Cer and LNnH-Cer [17], suggesting that residue 4 could be the key factor within the first block. Residue 4 lies at a second binding site, at the lateral side of the toxin, approximately 10 Å away from the primary binding site [18,19]. This patch was recently shown to bind blood group antigens and derivatives from human milk with millimolar affinities, involving residues 3, 4, 7, 18, 46, 47, and 94 [18,19,20,21,22,23]. Cross-talk between the two binding sites has been hypothesized earlier [10,24,25,26], and may be mediated through hydrogen-bonding networks connecting the two sites, and through the N-terminal α-helix. Recent NMR studies of hLTB identified residues with chemical shift changes upon titration with LNnT, supporting this hypothesis [27]. The largest chemical shift perturbations were seen for the backbones of Gln61 and the 50s loop, Gly33 and Asn14 (in order of the magnitude of the change; His13 could not be assigned), all located near the primary binding site. In addition, smaller chemical shift changes were observed for residues located between the two binding sites: 16, 38, 30, 96, and 8 (in order of the magnitude of the change), strongly suggesting cross-talk between the sites.

In order to test which residues are important for the binding of *N*-acetyllactosamine-terminated ligands to the primary binding site, we generated 10 single-site variants of pLTB towards residues of CTB or hLTB (Figure 1), and probed these for binding to the tetrasaccharide LNnT. In addition, we generated the hLTB variant H13R. A similar construct was previously tested for binding of the branched *N*-acetyllactosamine-terminated glycosphingolipid LNnH-Cer, in microtiter well assays [10]. Here, we probed the binding of both pLTB R13H and hLTB H13R for binding to linear LNnT, and the binding of pLTB R13H to branched LNnH, using surface plasmon resonance (SPR) spectroscopy. We further solved the crystal structure of wild-type (wt) pLTB in complex with LNnH to 1.45 Å resolution and compared it to the pLTB-LNnT structure (PDB ID: 2XRS [10]), revealing a comprehensive picture of the molecular interactions. 

## 2. Results

### 2.1. Quality Control

In order to verify proper folding of the toxin variants, we subjected all proteins to circular dichroism (CD) spectroscopy (Appendix A). In addition, several protein variants were tested for binding to the GM1 pentasaccharide to confirm that the primary binding site was intact, even when binding to LNnT was compromised. Wild-type pLTB and variant T4N bound with similar affinity to the GM1 pentasaccharide as previously reported, pLTB variant N94H bound more strongly, whereas S95A had a slightly lower affinity, but still in the nanomolar range (Table 1). The most notable change in GM1 binding affinity was observed for pLTB variant I58A, which dropped two orders of magnitude compared to wt pLTB, to micromolar affinity.

### 2.2. Surface Plasmon Resonance Spectroscopy with Lacto-N-Neotetraose

To date, most of the binding studies with *N*-acetyllactosamine-terminated ligands were carried out with microtiter well binding assays, revealing differences in multivalent binding, or avidity. In this work, we applied surface plasmon resonance (SPR) spectroscopy, in order to detect differences in monovalent binding. Each protein was immobilized on a sensor chip, and the binding to the soluble tetrasaccharide LNnT was measured. LNnT was purchased four times, here referred to as batches 1 to 4. In agreement with previous experiments using glycosphingolipids, LNnT has slightly higher affinity for pLTB than for hLTB, and poor affinity for CTB (Table 1; selected sensorgrams are shown in Figure 2). 

In the following paragraphs, we describe the results for the different toxin variants (Table 1 and Table 2), starting from the membrane-facing primary binding site, via residues connecting the primary and secondary toxin binding sites, to the secondary binding site, for blood group antigens. The final substitution discussed (residue 102) is at the B-subunit interface at the top of the B-pentamer (Figure 1a), where the A-subunit is positioned.

#### 2.2.1. Primary Binding Site Residues 13 and 58

The only residues differing between pLTB and hLTB are at positions 4, 13, 46, and 102 (Table 2). Of these, residue 13 is the only residue situated in the primary binding site. It was previously suggested that Arg13 is the main cause for the difference in binding patterns [16,28], a hypothesis subsequently supported by experimental findings that the hLTB variant H13R bound as strongly as pLTB to branched LNnH-Cer in microtiter well assays [10]. However, when we tested the variants pLTB R13H and hLTB H13R, we measured only a small difference in affinity to the linear tetrasaccharide LNnT by SPR spectroscopy compared to the wild-type proteins (Table 1). This fits with the observation that there are no physical interactions between residue 13 and the ligand in the crystal structure of pLTB in complex with LNnT (PDB ID: 2XRS [10]). Together, this suggests that residue 13 does not significantly affect the binding to soluble, linear *N*-acetyllactosamine-terminated structures.

Residue 58 is a primary-site residue located in the flexible 50s loop at the base of the toxin (Figure 1) that is identical in all three toxins. This residue was predicted to be important for the binding of LNnT by interactions between the Ile58 side chain and the *N*-acetyl group of GlcNAc [10]. To verify that LNnT only binds to the primary site in pLTB, and not to the secondary binding site, we mutated Ile58 to Ala, which has a shorter side chain. The I58A mutation abolished the binding to LNnT (Table 1 and Table 2), strengthening this hypothesis. This substitution also strongly affected binding to GM1, as discussed in Section 2.1. These results further suggest the importance of an entropic effect for ligand binding to the primary site, as predicted [10], with largest implications for weak binders like *N*-acetyllactosamine-terminating glycosphingolipids.

#### 2.2.2. Residues 94 and 95

Residues 94 and 95 are located between the two binding sites, with residue 95 positioned closer to the primary binding site than residue 94 (Figure 1). In LTB, the side chain of Ser95 engages in a hydrogen bond to Glu51 (PDB ID: 2XRS and 2XRQ [10]), an interaction that cannot be formed by CTB-specific Ala95 due to its missing hydroxyl group. Glu51 directly interacts with the terminal galactose of LNnT (Figure 1c), and further forms H-bonds to Gln56 and Lys91, which also binds the terminal galactose. In addition, Ser95 engages in a water-mediated bond to the backbone of Ile96 (not shown). Together, these features contribute to stability at the primary binding site. The side chain of residue 94 stretches into the secondary binding site, where it is involved in a water-mediated binding network also including the side chains of residues 16 and 18 [18,19] (Figure 1d) as well as Asn89, which via Asn14 and Asn90 links back to the primary binding site (Figure 1c). Residues 94 and 95 have previously only been mutated in combination, with LTB-specific residues (Asn94 and Ser95) correlating with stronger binding to *N*-acetyllactosamine-terminated glycosphingolipids compared to CTB residues (His94 and Ala95) [14]. In the SPR experiments presented here, both residues were proven to have an effect on the binding affinity, with N94H showing the stronger effect (Table 1 and Table 2).

#### 2.2.3. Secondary Binding Site Residues 4, 7, and 18

Intriguingly, binding to *N*-acetyllactosamine-terminated structures was found to be synergistically enhanced when LTB-specific residues Asn94 and Ser95 were introduced together with hLTB residues 1–25 in an LTB/CTB chimera called LCTB*H* [14,17]. This sequence contains important determinants of the secondary toxin binding site, which have already been in the focus of previous studies [10,17,18,20,27]. In particular, residue 4 was attributed major significance since back-substitution of LTB-specific Ser4 to CTB-specific Asn4 (creating toxin variant LCTB*K*) resulted in a complete loss of the chimera’s favourable binding properties [17].

In the crystal structures of the three toxins, residues 4 and 7 are some of the few amino acids that show side chain deviations, leading to differences in H-bonding patterns. They are positioned at two sides of a hairpin loop at the lateral side of the toxin, in the secondary binding site (Figure 1d). An α-helix stretches from residues 4 and 7 to residues 13 and 14, linking the secondary to the primary binding site (Figure 1b). In contrast to previous data, we could not observe any significant change in affinity upon substituting pLTB-specific Thr4 to either hLTB-specific residue Ser4 or CTB-specific Asn4, whereas substitution of residue 7 (E7D) resulted in decreased binding affinity (Table 1 and Table 2).

Tyr18 is positioned at the opposite side of the secondary binding site, H-bonding to residues 94 and 16, situated closer to the primary binding site (Figure 1d). His18 can maintain a similar, but not identical hydrogen-bonding pattern as Tyr18. We saw slightly weaker binding (*K*_D_ = 10.7 mM versus 9.4 mM) for the His18 variant, however, the difference is small and at the border of the precision limits of the measurements (Table 1).

#### 2.2.4. pLTB-Specific Residues 46 and 102

Residues 46 and 102 differ between hLTB and pLTB. Residue 46 is positioned at the top center of the secondary binding site, and residue 102 is located even further away from the primary binding site, close to the surface interacting with the A-subunit (Figure 1b). Neither the substitution of Glu46 to Ala (E46A) nor Lys102 to Glu (K102E) had any effect on the binding affinity to LNnT in our experiments (Table 1 and Table 2).

### 2.3. Surface Plasmon Resonance Spectroscopy with Lacto-N-Neohexaose

hLTB variant H13R was previously shown to enhance binding to branched LnNH-Cer to pLTB wild-type levels in microtiter well assays, strongly suggesting that residue 13 is the cause of the difference between the two toxins [10]. We therefore set out to test residue 13 variants (pLTB and hLTB) for differences in monovalent binding. However, we only found small differences in binding affinity that were statistically not significant compared to the variation we observed between analyte batches (Table 1). Moreover, due to the high costs of the analyte, we only performed few experiments. The differences observed are not sufficient to explain the previous results from microtiter well assays. 

### 2.4. Crystal Structures of pLTB wt with Lacto-N-Neohexaose

Given the inconclusive results from the SPR analysis presented in Section 2.3, we decided to determine the crystal structure of pLTB in complex with branched LNnH. The structure was obtained after co-crystallization and refined to a resolution of 1.45 Å (*R*/*R*_free_ = 17.5/20.1; Figure 3, Table 3). The crystal contained two B-pentamers in the asymmetric unit, giving us access to 10 crystallographically distinct B-subunits. The B-pentamers are positioned “top-to-top”, with the primary binding sites on opposite ends, similarly to recent LTB and CTB structures [20,26] (Figure 3a). The ligand LNnH is present in two of the 10 primary binding sites, on opposite ends of the decamer, and characterized by high-quality electron density (Figure 3a,b). Ligand binding stabilized the 50s loop (residues 51 to 60), which is disordered in all other subunits. LNnH binds with the β3-branch in the galactose pocket, as predicted [16], and superimposes well with the 1.8 Å crystal structure of pLTB in complex with LNnT (PDB ID: 2XRS [10]) (Figure 3d), whereas for GM1 (PDB ID: 2XRQ [10]), only the terminal Gal superimposes well with *N*-acetyllactosamine-terminated ligands (Figure 3e). The β6-branch is folded back on the reducing end of the sugar, with the β6-glycosidic bond interacting with Arg13 (Figure 3c; Table 4). The terminal galactose of the β6-branch is exposed, explaining why substitution can occur at the 2-, 3-, or 4-position [28]. Both ligand binding sites are in close proximity to neighboring molecules in the crystal, with similar contacts in both cases, therefore we cannot exclude that the β6-branch may adopt a different conformation (or be less ordered) in solution.

## 3. Discussion

We set out to explore the broader specificity of *E. coli* heat-labile enterotoxin (LT) compared to the cholera toxin (CT) with single-residue substitutions, using SPR spectroscopy. Unexpectedly, we recorded relatively large variation between analyte batches. Since sugars unlike proteins and nucleic acids do not contain any residues amenable for calibration using extinction coefficients, and the molecules are prohibitively expensive to obtain large amounts, we dealt with the variation by collecting as many data as possible per batch and including pLTB wt whenever possible as reference to enable comparison within batches of analytes. 

Most of the single-site variants showed no or very small differences in affinity to LNnT compared to the parent toxins. The largest drop in activity was recorded upon mutation of Ile58 (>50 mM), a residue present in all enterotoxins included in this study. Almost as dramatic was the effect of mutating Asn94 to His (>40 mM), which dropped the affinity to levels as low as for CTB. Given that pLTB and hLTB feature an asparagine residue at this position, it is likely that this substitution critically determines the lower monovalent affinity of CT versus LT. In addition, we observed decreases in binding affinity for pLTB variants S95A and E7D (15–20 mM), which is in a similar range compared to hLTB. Both of these residues are located between the primary and secondary toxin binding sites, with residue 95 closer to the primary binding site and residue 7 bordering the secondary binding site (Figure 1). A small reduction in binding affinity was also recorded for the Y18H variant (11 mM), while the other mutations showed no significant effects. Like Asn94, Tyr18 is located near the secondary toxin binding site, involved in the same H-bonding network as Asn94 (Figure 1c,d). The results may be explained by a change in the hydrogen-bonding network between the primary and secondary binding sites.

As notable as the effects of residues 94, 95, and 7 was the absence of significant effects for substitutions at positions 4 and 13. These two residues have received prime attention in earlier studies, using microtiter well assays [10,17]. It is worth noting that the SPR analysis performed here probed for monovalent binding, since the proteins were coupled to the SPR chips, whereas the microtiter well assays measured multivalent binding and avidity to immobilized glycolipids. The latter method gives a better view of the situation in vivo, however, both studies are needed to fully comprehend the underlying molecular interactions. Indications are that Ser/Thr4 and Arg13 are more important for multivalent binding than for monovalent binding. In the right context, Arg13 does not even seem be required for multivalent binding, as the CT-like toxin from *Citrobacter freundii* has been shown to bind even more strongly to LNnT-Cer than pLTB, despite exhibiting a histidine residue at position 13 [15]. Notably, this toxin features Glu7, Tyr18, Asn94, and Ser95—all residues identified as important in the current study. In this context, even Asn4 does not abrogate binding, in contrast to previous reports for the LTB/CTB chimera LCTB*K* compared to LCTB*H* [17].

Whereas the largest differences in binding affinity are between LTB and CTB, there are also small, but noticeable differences between pLTB and hLTB, accounting approximately for a factor two in monovalent binding affinity (Table 1). The only sequence differences between these two toxins concern residues 4, 13, 46, and 102, none of which showed a significant effect in our investigation. This strongly suggests that the substitutions have synergistic effects, as previously shown for regions 1–25 and 94–95 [14]. While we cannot be certain about the effects, we noticed that both residues 4 and 102 lie at the B-subunit interface and may affect binding affinity in concert, either by conformational pre-alignment or by pre-stabilizing the structure of the toxin. In this context it is interesting to note that pLTB crystallizes more easily compared to hLTB (this is even true for pLTB R13H [29]), which may rely on a similar effect, and points to pre-stabilization as a possible factor, reducing the entropic barrier to binding.

The original motivation for dissecting the broader binding specificity of LT versus CT was vaccine design [30]. The CT B-pentamer is a component of the major cholera vaccine [31], and the binding of CTB or LTB to cells has been shown to enhance immunogenicity [32,33]. It is conceivable that the inclusion of LTB or CTB/LTB hybrids with broader binding specificities could make vaccines more effective to combat several types of enterotoxigenic infections. This strategy is currently probed in clinical trials with the second generation oral ETEC vaccine [34,35], which includes a CTB/LTB chimera as well as a double-mutated LT holotoxin [36]. The work presented here may further aid the development of improved cholera and ETEC vaccines.

## 4. Materials and Methods 

### 4.1. Generation of Single-Site Variants

Nucleotide sequences of pLTB (Uniprot accession number P32890) and hLTB (Uniprot accession number P0CK94) were ordered from the GeneArt gene synthesis service (Thermo Fisher Scientific, Waltham, MA, USA), codon-optimized for expression in *E. coli*. The genes were subcloned into vector pET21b(+) (Novagen, Merck, Darmstadt, Germany). Single-site mutations were introduced using the Quikchange kit (Agilent Technologies, Santa Clara, CA, USA) or the NEB Q5 site-directed mutagenesis kit (New England Biolabs, Ipswich, MA, USA). The resulting plasmids were verified by DNA sequencing, and the purified proteins were later checked for proper folding using circular dichroism. The protein sequences for the proteins used in this study were as follows: pLTB: APQTITELCS EYRNTQIYTI NDKILSYTES MAGKREMVII TFKSGETFQV EVPGSQHIDS QKKAIERMKD TLRITYLTET KIDKLCVWNN KTPNSIAAIS MKN; hLTB: APQSITELCS EYHNTQIYTI NDKILSYTES MAGKREMVII TFKSGATFQV EVPGSQHIDS QKKAIERMKD TLRITYLTET KIDKLCVWNN KTPNSIAAIS MEN; and CTB (El Tor biotype): TPQNITDLCA EYHNTQIYTL NDKIFSYTES LAGKREMAII TFKNGAIFQV EVPGSQHIDS QKKAIERMKD TLRIAYLTEA KVEKLCVWNN KTPHAIAAIS MAN.

### 4.2. Production and Purification of Protein

pLTB variants were transformed into *E. coli* BL21 (DE3) cells for expression. Note that this expression system differs from the one used in previous studies from our group. The cells were grown in LB medium supplemented with 0.1 mg/mL ampicillin at 37 °C until an OD_600nm_ of 0.5 was reached. The temperature was lowered to 25 °C, the cells were induced with 0.5 mM IPTG, and incubated for 16–20 h. Cells were harvested by centrifugation at 6900× *g*, and the pellet re-suspended in periplasmic extraction buffer (5 mM MgCl_2_, 0.1 mg/mL lysozyme, protease inhibitor cocktail (Roche, Basel, Switzerland). The periplasmic fraction was separated from the cell debris by centrifugation, and dialyzed against PBS, before being applied to a d-Gal-sepharose gravity column (Thermo Fisher Scientific, Waltham, MA, USA). The bound protein was eluted with 300 mM d-Gal in PBS, and concentrated to 2–5 mg/mL. The protein was then applied to a Superdex75 size-exclusion chromatography column (GE Healthcare, Chicago, IL, USA), where the buffer was exchanged to 20 mM Tris-HCl pH 7.5, 100 mM NaCl. This protocol resulted in a yield of approximately 0.4 mg of purified protein per liter of culture. The proteins were kept at 4 °C for short term storage, and at −80 °C for long term storage.

hLTB wt and CTB were produced in *Vibrio* sp. 60, which secretes proteins into the growth medium. Cells were grown in LB medium supplemented with 15 g/L NaCl and 0.1 mg/mL ampicillin at 30 °C until an OD_600nm_ of 0.2 was reached. After induction with 0.5 mM IPTG, expression proceeded for 16–24 h until the cells were separated from the medium by centrifugation at 40,000× *g*. The cleared medium was applied to a d-Gal-sepharose gravity column, and eluted with 300 mM d-Gal in PBS. The protein was subsequently concentrated to 2–5 mg/mL and applied to a Superdex75 size-exclusion column, where the buffer was exchanged to 20 mM Tris-HCl pH 7.5, 100 mM NaCl. This protocol resulted in a yield of approximately 6 mg of purified protein per liter of culture. The proteins were kept at 4 °C for short term storage, and at −80 °C for long term storage. 

### 4.3. Analysis by Surface Plasmon Resonance

Prior to SPR experiments, the protein was dialyzed against PBS. The experiments were performed on a Biacore T100 biosensor system (GE Healthcare, Chicago, IL, USA) at the Infrastructural Center for Analysis of Molecular Interactions, University of Ljubljana, Slovenia. All experiments were carried out at 25 °C in HBS-EP running buffer (10 mM Hepes pH 7.4, 150 mM NaCl, 3 mM EDTA, 0.005% (*v*/*v*) surfactant P20), and the analytes lacto-*N*-neotetraose (LNnT; Elicityl-oligotech, Crolles, France, product code GLY021) or branched lacto-*N*-neohexaose (LNnH; Dextra Laboratories LtD, Reading, UK, product code L605) were solubilized in the same buffer. LnNT was ordered and shipped from the supplier at four different time points (batches 1 to 4). The analytes were weighed out using a high precision scale, but inaccuracies may have been introduced at this stage due to the difficulties in measuring a few milligrams of material. This or differences in additives like salts or impurities are most likely the cause of the small discrepancies between batches. The proteins were diluted in 10 mM sodium acetate pH 5.5 and immobilized by amine coupling to a CM5 sensor chip (GE Healthcare, Chicago, IL, USA) to a response of 2500–6000 RU. 

The experiments with LNnT were done in four rounds: the first round included pLTB R13H and hLTB H13R (without the wild types since the values were thought to be known from the preliminary experiments). LNnT was injected over the surfaces with a flow rate of 5 µL/min, 30 s contact time at increasing concentrations of the carbohydrate analyte (0.156 mM to 40 mM), with one injection per concentration, and repeated two times. The second round included pLTB wt, E7D, I58A, and N94H, with a flow rate of 20 µL/min for 60 s (followed by a 60 s dissociation phase and allowing 60 s of stabilization prior to the next injection), and repeated three times. The third round included variants pLTB wt, T4N, T4S, Y18H, E46A, S95A, K102E, and CTB, with a similar protocol, with an increasing analyte concentration up to 30 mM. Due to a slight increase in baseline, 3 mM NaOH was used for regeneration of the chip between each injection. In the fourth round we repeated pLTB wt and T4N, using the same protocol as previously. The experiments with LNnH were performed equivalently to those with LNnT with a few exceptions. The experiments were done once with two runs of analyte, due to the high cost of the analyte. LNnH was injected over the surfaces at increasing concentrations of analyte (0.0098 mM to 10 mM), with one injection per concentration. 

The SPR experiments with analyte GM1 pentasaccharide (GM1a; Elicityl-oligotech, Crolles, France, product code GLY096) were performed as described in [26]. In brief, the analyte was injected at increasing concentrations (up to 200 nM) over the CM5 sensor chip to which the protein was immobilized. All experiments were performed in duplicate. The dissociation constants for LNnT and LNnH were calculated by using a steady-state affinity model, while the GM1a data was fitted to a Langmuir 1:1 interaction model, using the Biacore T100 evaluation software. 

### 4.4. Crystallographic Analysis

pLTB (5.6 mg/mL) and LNnH were mixed at a molar ratio of 1:10 (B-subunit:ligand) 2 h prior to crystallization. Sitting drop vapor diffusion experiments (300 nL protein solution + 300 nL crystallization buffer) were set up using an Oryx4 crystallization robot (Douglas Instruments, East Garston, UK). pLTB in complex with LNnH crystallized at 20 °C in the PACT premier screen condition A9 (0.1 M sodium acetate pH 5.0, 20% *w*/*v* PEG 6000, 0.2 M LiCl). Diffraction-quality crystals were flash-cooled in a nitrogen cryo-stream.

Synchrotron data collection was performed at beamline ID23-2, ESRF, Grenoble, France (100 K, 0.8729 Å). Data were processed with *xia*2/*DIALS* and *AIMLESS* from the *CCP*4 software suite [37,38], and cut to a resolution of 1.65 Å by assessing statistical parameters including the *CC*_1/2_ value. The structure was solved by molecular replacement with the program *Phaser* [39] from the *CCP*4 software suite using PDB entry 1DJR [40] as the search model. To reduce model bias, five cycles of refinement including two cycles with simulated annealing were carried out with the Phenix software suite [41]. The structure was refined by alternating manual building with *Coot* and automatic refinement with *REFMAC*5 [42,43]. Refinement steps involved local non-crystallographic symmetry (NCS) restraints and TLS model parameterization (*REFMAC*5, automatic, five cycles). *PDB_REDO* was used to evaluate and automatically optimize the model [44]. Water molecules were first automatically placed in *Coot* and then manually inspected. LNnH was included last and built using MAKE LIGAND (*AceDRG* [45]) from the *CCP*4 software suite and an isomeric SMILES string (from PubChem Sketcher). The disulphide bond between Cys9 and Cys96 showed signs of being partially reduced in all subunits, and was modeled with an alternative conformation by changing the SSBOND entries in the PDB file to LINKR entries. Data initially processed with *DIALS* (without scaling by *xia*2) were directly scaled with *AIMLESS* leading to better data statistics. Data were cut to a resolution of 1.45 Å by assessing statistical parameters including the *CC*_1/2_ value, and used for final refinement steps. The final model was analysed using the Analyse geometry task of the *CCP*4 software suite. The atomic coordinates and structure factors have been deposited in the PDB under entry 6IAL. Figures were generated using PyMol (Schrödinger LLC, New York City, NY, USA).

## 5. Conclusions

Using SPR spectroscopy, we probed the effect of a number of single-site substitutions between pLTB, hLTB, and CTB. In addition to Ile 58, which is conserved in all toxin variants, the largest effect was found upon substituting Asn94, followed by Ser95 and Glu7, and Tyr18. Intriguingly, these residues lie on two paths connecting the primary and secondary toxin binding sites, which have previously been implicated in allosteric cross-talk [10,26,27]. This cross-talk may not only be important for the communication between the sites, but also directly affect binding affinity and specificity, with important implications for the biological mechanism of the toxins. 

## Figures and Tables

**Figure 1 ijms-20-00703-f001:**
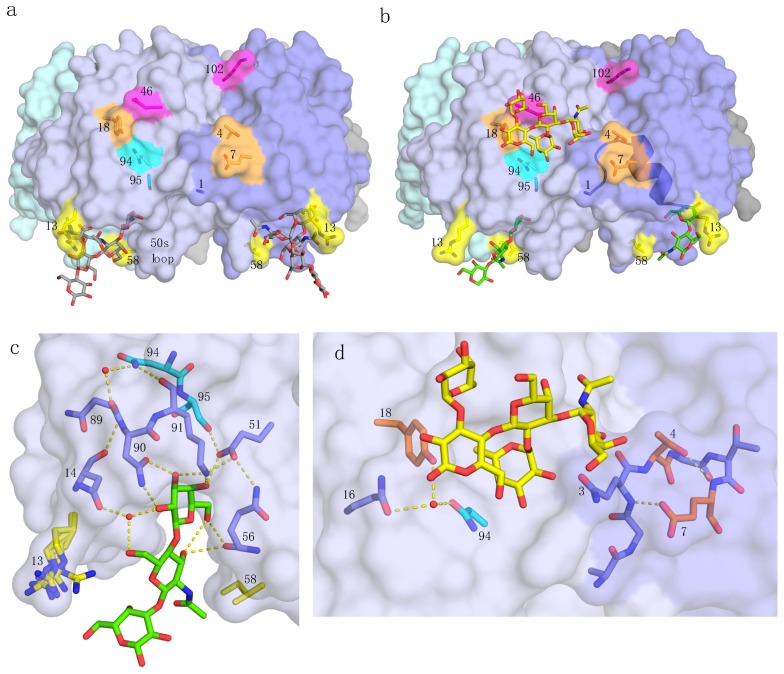
pLTB structure and interactions. Residues and features of special importance for this work are marked, with residues colored in groups, as discussed in Section 2. (**a**) pLTB (PDB ID: 2XRQ [10]) shown in surface representation, colored by subunit. The GM1 oligosaccharide is shown in stick representation, with grey carbons. (**b**) pLTB in complex with lacto-*N*-neotetraose (LNnT; PDB ID: 2XRS [10]), in a collage with an analogue of the blood group A-pentasaccharide bound to the secondary binding site, superimposed from hLTB structure 2O2L [19]. Not all of the carbohydrate residues were modeled for LNnT, due to weak electron density at the non-reducing end. The α-helix connecting residues 1 through 14 is shown in cartoon representation. (**c**) Close-up view of the primary binding site of pLTB with LNnT (PDB ID: 2XRS [10]). Important residues are shown in stick representation, and the hydrogen-bonding network in yellow dotted lines, extending up to residue 94 (cyan). Two conserved water molecules are depicted as red spheres. Residue 13 (yellow) adopts alternative conformations in the different subunits, and residue 58 (yellow) has van der Waals contacts to the methyl group of GlcNAc. (**d**) Close-up view of relevant residues at the secondary binding site of pLTB (PDB ID: 2XRS [10]), with the A-pentasaccharide (from PDB ID: 2O2L [19]) superimposed in yellow sticks. Important residues on both sides of the secondary binding site are shown.

**Figure 2 ijms-20-00703-f002:**
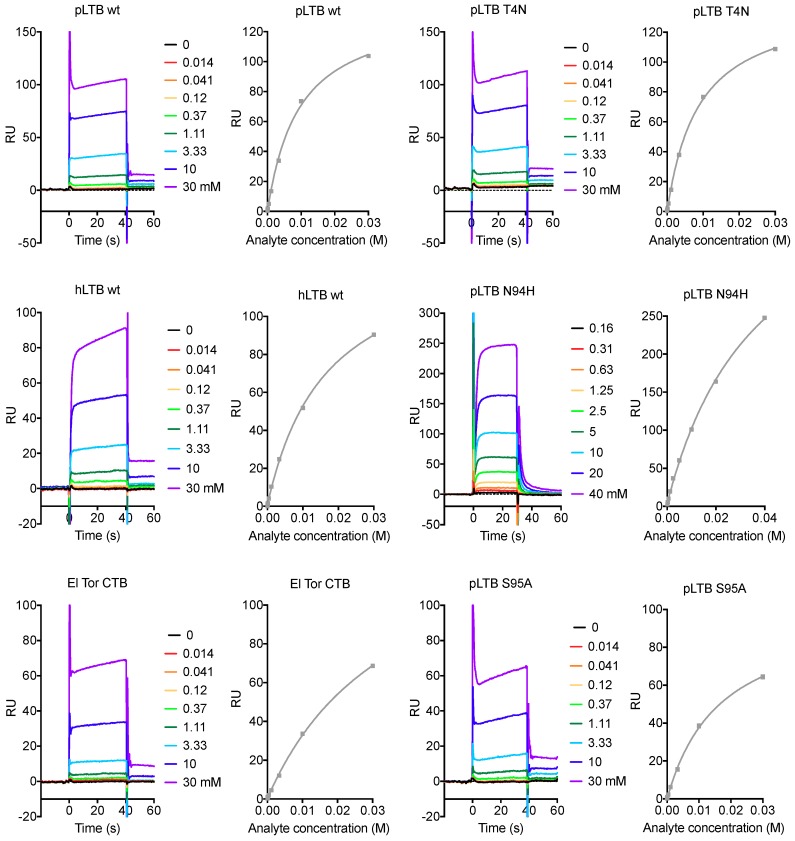
Selected SPR sensorgrams and affinity plots for analyte LNnT. The colored graphs are responses to increasing concentrations of the analyte LNnT, in multi-steady state affinity experiments (response units (RU) versus analyte concentration). The resulting steady-state values are plotted as dots with a fitted curve, using the Biacore T100 evaluation software. The responses are dependent on the protein immobilization rates, therefore the RU-axes are not comparable between protein variants.

**Figure 3 ijms-20-00703-f003:**
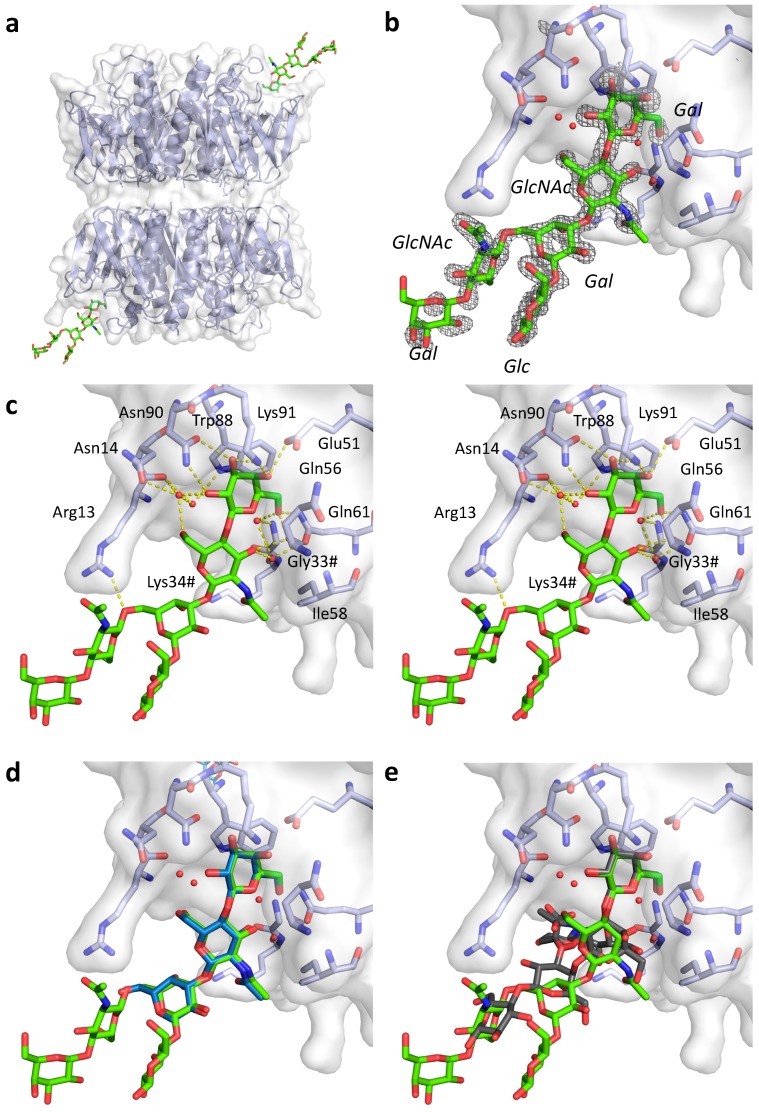
Structure of pLTB in complex with lacto-*N*-neohexaose (PDB ID: 6IAL, this work; LNnH shown in stick representation with green carbons). (**a**) Overview of the asymmetric unit, with two B-pentamers positioned “top-to-top” and two LNnH molecules bound. (**b**) Close-up view of the ligand binding site, with σ_A_–weighted *F*_o_-*F*_c_ electron density map shown in grey mesh contoured at 3.0 σ, generated before placing the ligand. (**c**) Stereo-image of the binding site, with important residues labeled, H-bonding interactions shown as yellow dotted lines, and selected water molecules depicted as red spheres. Residues from the neighboring subunit are indicated by a hash (#). (**d**) Close-up view of the LNnH binding site superimposed with LNnT (blue) from structure 2XRS [10]. (**e**) Close-up view of the LNnH binding site superimposed with GM1 pentasaccharide (grey) from structure 2XRQ [10].

**Table 1 ijms-20-00703-t001:** *K*_D_ values measured by SPR, grouped by analyte and batch number

Protein Variant	GM1a	LNnT Batch 1	LNnT Batch 2	LNnT Batch 3	LNnT Batch 4	LNnH
pLTB wt	37 ± 0.5 nM	6.7 ± 0.5 mM	-	9.4 ± 0.1 mM	8.0 ± 0.6 mM	5 ± 1 mM
pLTB T4N	32 ± 0.3 nM	-	-	8.6 ± 0.1 mM	7.8 ± 0.7 mM	-
pLTB T4S	-	-	-	8.9 ± 0.4 mM	-	-
pLTB E7D	-	-	12 ± 3 mM *	-	-	-
pLTB R13H	-	9.3 ± 1.7 mM	-	-	-	7 ± 1 mM
hLTB H13R	-	11.5 ± 0.2 mM	-	-	-	-
pLTB Y18H	-	-	-	10.7 ± 0.1 mM	-	-
pLTB E46A	-	-	-	9.8 ± 0.1 mM	-	-
pLTB I58A	2.3 ± 1.7 μM	-	n.b.	-	-	-
pLTB N94H	23 ± 1 nM	-	>40 mM **	-	-	-
pLTB S95A	65 ± 4 nM	-	-	18 ± 1.4 mM	-	-
pLTB K102E	34 ± 0.8 nM	-	-	9.7 ± 0.5 mM	-	-
hLTB wt	-	-	-	16 ± 0.03 mM	-	-
CTB	-	-	-	>36 mM **	-	-

* For variant E7D, four titrations were carried out, giving *K*_D_ values of >27 mM, 11.5 mM, 9.8 mM, and 15.3 mM. The reported *K*_D_ does not include the outlier. ** Accurate value impossible to determine since they are outside of the concentration range of LNnT analyte (30/40 mM) used in this study.

**Table 2 ijms-20-00703-t002:** Sequence and affinity differences between pLTB, hLTB, CTB, and toxin variants.

Residue	pLTB	hLTB	CTB *	Toxin Variant	Effect on LNnT Affinity
1	Ala	Ala	Thr *		
4 **	Thr	Ser	Asn	pLTB T4N/T4S	Like pLTB wt
7	Glu	Glu	Asp	pLTB E7D	Lower affinity
10	Ser	Ser	Ala	-	-
13	Arg	His	His	pLTB R13H	Similar to pLTB wt
				hLTB H13R	Similar to hLTB wt
18	Tyr	Tyr	Tyr *	pLTB Y18H	Slightly lower affinity
20	Ile	Ile	Leu *	-	-
25	Leu	Leu	Phe	-	-
31	Met	Met	Leu	-	-
38	Val	Val	Ala	-	-
44	Ser	Ser	Asn	-	-
46	Glu	Ala	Ala	pLTB E46A	Like pLTB wt
47	Thr	Thr	Ile *	-	-
58	Ile	Ile	Ile	pLTB I58A	No binding
75	Thr	Thr	Ala	-	-
80	Thr	Thr	Ala	-	-
82	Ile	Ile	Val	-	-
83	Asp	Asp	Glu	-	-
94	Asn	Asn	His	pLTB N94H	Lower affinity
95	Ser	Ser	Ala	pLTB S95A	Lower affinity
102	Lys	Glu	Ala	pLTB K102E	Like pLTB wt

* There are two major *V. cholerae* biotypes, classical, and El Tor. Here, we studied the El Tor variant, which differs from classical CTB in residues 18 (H18Y) and 47 (T47I). Compared to recombinant CTB used in previous studies, also residues 1 (A1T) and 20 (I20L) differ [14,17]. Since pLTB substitution A1T was discussed in previous work [10], we also performed preliminary experiments for this variant, which showed a similar affinity compared to wt pLTB, and thus seems to be of minor importance for LNnT binding. ** Color coding of residues according to Figure 1a,b and Section 2.2.1, Section 2.2.2, Section 2.2.3 and Section 2.2.4. (yellow, primary binding site, discussed in Section 2.2.1; cyan, between primary and secondary binding site, Section 2.2.2; orange, secondary binding site, Section 2.2.3; magenta, residues discussed in Section 2.2.4).

**Table 3 ijms-20-00703-t003:** Data collection and refinement statistics

Protein	pLTB + LNnH
PDB ID	6IAL
**Data collection**	
Space group	*P*2_1_
**Cell dimensions**	
a, b, c (Å)	77.1, 65.6, 96.3
β (°)	108.6
Resolution (Å)	68.7–1.45 * (1.47–1.45) **
No. of unique reflections	160,263 (7,848)
*CC*_(1/2)_ (%)	99.6 (45.8)
*R* _merge_	0.13 (1.15)
(*I*)/σ(*I*)	6.4 (1.3)
Multiplicity	4.4 (4.5)
Completeness (%)	99.6 (99.0)
**Refinement**	
R_cryst_/R_free_ (%)	17.5/20.1
No. of atoms	
Protein	8622
Ligand/ion	158/15
Water	644
Average *B*-factors (Å^2^)	
Protein	19.6
Ligand /ion	23.2/19.7
Water	23.2
r.m.s.d. bonds (Å)	0.01
r.m.s.d. angles (°)	1.7

* Data collected on a single crystal. ** Values for the highest resolution shell are shown in parentheses.

**Table 4 ijms-20-00703-t004:** Protein–carbohydrate interactions to LNnH (PDB ID: 6IAL)

Residue	Donor/acceptor	Distance (Å) Site 1	Distance (Å) Site 2
Arg13	NH1-O6 Galβ4 (second Gal)	2.9	3.2
Asn14	OD1-O2 Galβ3 via solvent	3.0–H_2_O–2.9	-
	and O6 GlcNAcβ3 via solvent	3.0–H_2_O–2.8	-
Gly33#	N-O6 Galβ4 via solvent	2.8–H_2_O–2.9	2.9–H_2_O–3.0
Glu51	OE2-O4 Galβ4	2.7	2.7
Gln56	O-O6 Galβ4	2.7	2.6
	O-O3 GlcNAcβ3	2.8	2.9
Ile58	GlcNAcβ3	3.6*	4.3 *
Gln61	NE2-O6 Galβ4	3.0	3.0
	OD1-O3 GlcNAcβ3 via solvent	2.8 – H_2_O – 2.8	2.9–H_2_O–2.9
Asn90	ND2-O2 Galβ4	2.9	2.9
	OD1-O3 Galβ4	2.9	2.9
Lys91	NZ-O3 Galβ4	2.8	2.8
	NZ-O4 Galβ4	2.9	2.8

* strong van der Waals interaction.

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
