# Peer review of "Specificity of Escherichia coli Heat-Labile Enterotoxin Investigated by Single-Site Mutagenesis and Crystallography"

_ijms, 2019, doi:10.3390/ijms20030703_

Round 1

Reviewer 1 Report

The authors have adequately responded to all of the comments and the paper is much improved.

Reviewer 2 Report

The submitted manuscript "Specificity of Escherichia coli heat-labile enterotoxin investigated by single-site mutagenesis and crystallography" should be accepted in its present form. The modifications by the authors to the original submission, based on previous comments from reviewers, add much clarity to the paper and have improved clarity of the manuscript by adding additional explanations where required.

This manuscript is a resubmission of an earlier submission. The following is a list of the peer review reports and author responses from that submission.

Round 1

Reviewer 1 Report

This well-written manuscript provides a detailed analysis of how various single-site mutations affect enterotoxin affinity, building on previous studies. The results are very interesting for the potential development of an LTB/CTB hybrid construct for enhanced immunogenicity of vaccines, and in addition is interesting for the improved understanding of lectin interactions in general. The lack of a complete data set for all protein/ligand combinations and the observed batch-to-batch LNnT variation does somewhat weaken the scientific soundness, but the batch variance is still acceptable and the trends and observations made are convincing; the authors address this deficiency in data due to financial constraints.

Specific comments:

-  Line 106, Figure 1 (b and c): In the crystal structure with lacto-N-neotetraose, only 3 of the 4 pyranose rings are illustrated in the structure (presumably due to a lack of electron density) – it would be more clear if this were stated in the figure description or displayed more clearly in the image that a region of the ligand has been omitted.

-  Line 123: “selected protein variants” were tested for GM1 affinity – the 3 examined appear very randomly selected, and it would be helpful to include a brief description of why these mutations were chosen. Some of the other mutants had quite drastic substitutions (e.g. E46A) and it is not obvious why these weren’t screened for an intact primary binding site as well.

-  Line 127: It is not clear what is different between the different batches, as this ligand was purchased from a commercial supplier. Different shipments from the supplier? Different stock solutions? It would be informative to include this in the methods.

-  Line 128: The first sentence “For variant A1T, preliminary data…” seems misplaced and is not referred to in the remainder of the manuscript. Should this be amended/removed?

-  Line 130: In bottom line of Table 1, the * symbol should be corrected to ** to correlate with the appropriate statement.

-  Line 152: in Table 2 headings, it would be useful to re-iterate (based on the information provided in Table 1) that the variants listed are pLTB mutants.

-  Line 155: The subheadings/grouping used throughout section 2.2 are very helpful for clarity, as it very nicely divides the different mutations into primary versus secondary binding site versus bridging region. This is really helpful for making sense of the list of mutations which in the tables are just listed in ascending sequence order.

-  Line 290: Without accessing the referenced manuscript, it is not clear what LCTBK and LCTBH are referring to, and it would be helpful to further define these in this manuscript.

-  Line 314: In the conclusion, the authors address the CT-like C. freundii toxin which, other than being very briefly (1-2 sentences) thrown-in at the end of a paragraph in the introduction, is not addressed throughout the manuscript. It would be useful to add something in the discussion which addresses the points raised in the conclusion, since as it stands it seems a very disconnected argument here.

Author Response

Response to Reviewer 1

Comments and Suggestions for Authors

This well-written manuscript provides a detailed analysis of how various single-site mutations affect enterotoxin affinity, building on previous studies. The results are very interesting for the potential development of an LTB/CTB hybrid construct for enhanced immunogenicity of vaccines, and in addition is interesting for the improved understanding of lectin interactions in general. The lack of a complete data set for all protein/ligand combinations and the observed batch-to-batch LNnT variation does somewhat weaken the scientific soundness, but the batch variance is still acceptable and the trends and observations made are convincing; the authors address this deficiency in data due to financial constraints.

Specific comments:

-  Line 106, Figure 1 (b and c): In the crystal structure with lacto-N-neotetraose, only 3 of the 4 pyranose rings are illustrated in the structure (presumably due to a lack of electron density) – it would be more clear if this were stated in the figure description or displayed more clearly in the image that a region of the ligand has been omitted.

Indeed, this could be confusing to our readers. We have now added the following explanatory sentence to the legend of Figure 1: “Not all of the carbohydrate residues were modelled for LNnT, due to weak electron density at the non-reducing end”.

-  Line 123: “selected protein variants” were tested for GM1 affinity – the 3 examined appear very randomly selected, and it would be helpful to include a brief description of why these mutations were chosen. Some of the other mutants had quite drastic substitutions (e.g. E46A) and it is not obvious why these weren’t screened for an intact primary binding site as well.

We chose protein variants with reduced binding affinity to LNnT. This has now been described more clearly: “…several protein variants were tested for binding to the GM1 pentasaccharide to confirm that the primary binding site was intact, even when binding to LNnT was compromised.” – In the revised manuscript, we further added data for two additional pLTB variants, N94H and I58H. Especially the latter was interesting, since it showed a two-fold decrease in binding affinity compared to wild-type pLTB. This supports the importance of this residue, in line with earlier predictions by Holmner et al., 2011. We have expanded on this discussion in Section 2.2.1.

-  Line 127: It is not clear what is different between the different batches, as this ligand was purchased from a commercial supplier. Different shipments from the supplier? Different stock solutions? It would be informative to include this in the methods.

As explained in response to Reviewer 1, they are different shipments from the supplier. We have now added explanatory sentences in the Results and Methods sections.

-  Line 128: The first sentence “For variant A1T, preliminary data…” seems misplaced and is not referred to in the remainder of the manuscript. Should this be amended/removed?

Since the A1T variant was discussed in a previous publication (Holmner et al., JMB 2011) and it was speculated that it may have an effect on LNnT binding affinity, we would like to mention these preliminary results. However, we agree that the placement is ill chosen and have instead added a comment to the reference of Table 2.

-  Line 130: In bottom line of Table 1, the * symbol should be corrected to ** to correlate with the appropriate statement.

Corrected.

-  Line 152: in Table 2 headings, it would be useful to re-iterate (based on the information provided in Table 1) that the variants listed are pLTB mutants.

Good idea. We have now included “pLTB”.

-  Line 155: The subheadings/grouping used throughout section 2.2 are very helpful for clarity, as it very nicely divides the different mutations into primary versus secondary binding site versus bridging region. This is really helpful for making sense of the list of mutations which in the tables are just listed in ascending sequence order.

To further improve readability, we have now colour-coded the residues in Table 2 according to Figure 1.

-  Line 290: Without accessing the referenced manuscript, it is not clear what LCTBK and LCTBH are referring to, and it would be helpful to further define these in this manuscript.

We agree that this was not well enough explained and now added more details to the Introduction (lines 84-85; corresponding to lines 90-91 in the document with tracked changes), and to Section 2.2.3 (line 213; or 271 when changes were tracked).

-  Line 314: In the conclusion, the authors address the CT-like C. freundii toxin which, other than being very briefly (1-2 sentences) thrown-in at the end of a paragraph in the introduction, is not addressed throughout the manuscript. It would be useful to add something in the discussion which addresses the points raised in the conclusion, since as it stands it seems a very disconnected argument here.

We removed reference to the CT-like toxin from C. freundii from the Conclusion to draw less attention to it. It was merely a case in point.

Reviewer 2 Report

The manuscript titled "Specificity of Escherichia coli heat-labile enterotoxin investigated by single-site mutagenesis" by Heggelund et al. describes the carbohydrate-binding specificity of a bacterial heat-labile enterotoxin using single-site mutagenesis in conjunction with surface plasmon resonance spectroscopy and X-ray crystallography. This work is logically presented and well described, and would be important to researchers in this field. However, several points described below should be addressed before publication.

Comment#1: Page 1, Line 2-, Title

Although the result of the X-ray crystallographic analysis provided conclusive evidence for the interaction of pLTB and LNnH, this information is not conveyed in the title of the manuscript. Therefore, the title should be reconsidered. For example, ‘……..single-site mutagenesis and X-ray crystallography’ might be more informative to the readers.

Comment#2: Page 1, Line 28

The relation of the words ‘multivalent binding’ and ‘enhancing avidity’ is not clear. It could be changed to ‘multivalent binding and enhancing avidity’ or simply ‘multivalent binding’ by deleting ‘enhancing avidity’.

Comment#3: Page 4, Line 127-, Table 1

The meaning of the term ‘batch’ in the table is unclear though the difference in affinity among the ‘batches’ of analyte is important as described in the first paragraph of the discussion. Considering the importance of the ‘batch’ of the analyte, the meaning and the importance of ‘batch’ should be described clearly, e.g. in the first paragraph of the section 2.2. The meanings of the empty cells in Table 1 should also be explained in the footnote of the table.

Comment#4: Page 4, line 139

The data presented in figure 2 are selected SPR sensorgrams as described in the title of the figure. This should also be described in the main text to avoid confusion, e.g. ‘Table 1, and selected sensorgrams were shown in Figure 2’ instead of ‘Table 1, Figure 2’.

Comment#5: Page 6, Line 152-, Table 2

The explanations about the contents of the table are obscure and might cause confusion in the readers.

1) The meanings of ‘CTB EI Tor’ and ‘CTB classical’ are not described and unfortunately, I could not determine what they mean and the difference in them even from other parts of this manuscript. Their meaning should be clearly described.

2) The term ‘variant’ might also cause confusion. What does ‘variant’ mean? To clarify this point, for e.g., ‘T4N/T4S’ should be replaced with ‘pLTB T4N/T4S’.

3) The description ‘Effect on LNnT’ should be replaced with ‘Effect of single-site mutagenesis on the affinity for LNnT’.

Comment#6: Page 7, Line 178-180

Unfortunately, I could not understand the meaning of the sentence ‘Glu51 further… in this region’. Please modify this sentence so that it is easily understood.

Comment#7: Page 7, Line 200

The description ‘we do not…’ might be an error; if so, it should be written as ‘we could not…’.

Comment#8: Page 11, Line 267

I could not get the meaning of ‘the toxins of both cholera biotypes’. Does ‘both’ mean pLTB and EI Tor CTB or EI Tor CTB and Classical CTB or otherwise? Please describe clearly.

Comment#9: Page 11, Line 288

The description ‘all the other…’ might be an error; if so, it should be written as ‘all these…’.

Comment#10: Page 13, Line 358-, Section 5.3

The description in this section is the same as that for supplemental figure 1. Therefore the contents of this section should be described in the supplemental method or supplemental figure in this version of this manuscript or should be moved into the main manuscript.

Comment#11

In addition to Comment#7 and #9, I found that the meanings of several sentences might be difficult to understand. Therefore, I strongly recommend the authors to carefully read through the manuscript before publication and correct these sentences.

Author Response

Response to Reviewer 2

Comments and Suggestions for Authors

The manuscript titled "Specificity of Escherichia coli heat-labile enterotoxin investigated by single-site mutagenesis" by Heggelund et al. describes the carbohydrate-binding specificity of a bacterial heat-labile enterotoxin using single-site mutagenesis in conjunction with surface plasmon resonance spectroscopy and X-ray crystallography. This work is logically presented and well described, and would be important to researchers in this field. However, several points described below should be addressed before publication.

Comment#1: Page 1, Line 2-, Title

Although the result of the X-ray crystallographic analysis provided conclusive evidence for the interaction of pLTB and LNnH, this information is not conveyed in the title of the manuscript. Therefore, the title should be reconsidered. For example, ‘……..single-site mutagenesis and X-ray crystallography’ might be more informative to the readers.

We appreciate the suggestion from the reviewer and have changed the title in a shortened form, ending with “single-site mutagenesis and crystallography”.

Comment#2: Page 1, Line 28

The relation of the words ‘multivalent binding’ and ‘enhancing avidity’ is not clear. It could be changed to ‘multivalent binding and enhancing avidity’ or simply ‘multivalent binding’ by deleting ‘enhancing avidity’.

Changed to “multivalent binding and avidity”.

Comment#3: Page 4, Line 127-, Table 1

The meaning of the term ‘batch’ in the table is unclear though the difference in affinity among the ‘batches’ of analyte is important as described in the first paragraph of the discussion. Considering the importance of the ‘batch’ of the analyte, the meaning and the importance of ‘batch’ should be described clearly, e.g. in the first paragraph of the section 2.2. The meanings of the empty cells in Table 1 should also be explained in the footnote of the table.

Explanations have been added to the Results and Methods sections. Results: “LNnT was purchased and shipped four times, here referred to as batches 1 to 4.” Methods: “LnNT was ordered and shipped from the supplier at four different time points (batches 1 to 4).”

Comment#4: Page 4, line 139

The data presented in figure 2 are selected SPR sensorgrams as described in the title of the figure. This should also be described in the main text to avoid confusion, e.g. ‘Table 1, and selected sensorgrams were shown in Figure 2’ instead of ‘Table 1, Figure 2’.

We appreciate this very good suggestion from Reviewer 1 and have implemented the change.

Comment#5: Page 6, Line 152-, Table 2

The explanations about the contents of the table are obscure and might cause confusion in the readers.

1) The meanings of ‘CTB EI Tor’ and ‘CTB classical’ are not described and unfortunately, I could not determine what they mean and the difference in them even from other parts of this manuscript. Their meaning should be clearly described.

Indeed, we forgot to clarify the difference between ‘CTB El Tor’ and ‘CTB classical’ and would like to apologize for that. There are two major V. cholerae biotypes, classical and El Tor. Here, we studied the El Tor variant, which differs from classical CTB in residues 18 (H18Y) and 47 (T47I). We have now explained this in the footnote to Table 2 and in the Methods section, but took away the specifications in the remainder of the text.

2) The term ‘variant’ might also cause confusion. What does ‘variant’ mean? To clarify this point, for e.g., ‘T4N/T4S’ should be replaced with ‘pLTB T4N/T4S’.

We have implemented the changes, as suggested by the reviewer. To clarify the term “variant”, which is the common term for a protein with sequence alterations, we changed it to “Toxin variant”.

3) The description ‘Effect on LNnT’ should be replaced with ‘Effect of single-site mutagenesis on the affinity for LNnT’.

To clarify the description, while at the same time keeping it short, we have now changed it to “Effect on LNnT affinity”.

Comment#6: Page 7, Line 178-180

Unfortunately, I could not understand the meaning of the sentence ‘Glu51 further… in this region’. Please modify this sentence so that it is easily understood.

We have modified this part to read: “Glu51 … further forms H-bonds to Gln56 and Lys91, which also binds the terminal galactose. In addition, Ser95 engages in a water-mediated bond to the backbone of Ile96 (not shown). Together, these features contribute to stability at the primary binding site.” We hope that has improved clarity.

Comment#7: Page 7, Line 200

The description ‘we do not…’ might be an error; if so, it should be written as ‘we could not…’.

Changed

Comment#8: Page 11, Line 267

I could not get the meaning of ‘the toxins of both cholera biotypes’. Does ‘both’ mean pLTB and EI Tor CTB or EI Tor CTB and Classical CTB or otherwise? Please describe clearly.

Indeed, this sentence is unnecessarily complicated and we have taken away the specifications of toxin biotypes.

Comment#9: Page 11, Line 288

The description ‘all the other…’ might be an error; if so, it should be written as ‘all these…’.

Good suggestion. Corrected.

Comment#10: Page 13, Line 358-, Section 5.3

The description in this section is the same as that for supplemental figure 1. Therefore the contents of this section should be described in the supplemental method or supplemental figure in this version of this manuscript or should be moved into the main manuscript.

Good point. We have moved the description to the Supplementary Information, as suggested.

Comment#11

In addition to Comment#7 and #9, I found that the meanings of several sentences might be difficult to understand. Therefore, I strongly recommend the authors to carefully read through the manuscript before publication and correct these sentences.

We have once more carefully gone through the manuscript and hope that it now is clearer.